# Factor Structure of the Polish Version of Multidimensional Body-Self Relations Questionnaire-Appearance Scales (MBSRQ-PL)

**DOI:** 10.3390/ijerph19106097

**Published:** 2022-05-17

**Authors:** Bernadetta Izydorczyk, Sebastian Lizińczyk

**Affiliations:** 1Institute of Psychology, Jagiellonian University, 30-348 Krakow, Poland; 2Faculty of Psychology, SWPS University of Social Sciences and Humanities, 40-326 Katowice, Poland; octans@wp.pl

**Keywords:** Multidimensional Body-Self Relations Questionnaire, adaptation, body image

## Abstract

In Poland, there is no tool for measuring the variable body image that would have standardization or Polish norms adjusted to the population of both women and men. The available Multidimensional Body-Self Relations Questionnaire (MBRSQ) validation developed in Poland was based on a small population of young women only. The aim of this article is to present Polish adaptation and standardization for polish adult women and men (MBSRQ-AS). In the study, the psychometric properties and factor structure of the Polish version of MBSRQ-AS were tested. The participants were 2688 people, including 1699 young women and 1089 young men. In order to investigate the psychometric properties of the adapted Polish tool, an exploratory factor analysis (EFA) was performed. Then, the reliability coefficients were calculated, and the descriptive statistics of individual subscales were checked. The separated subscales are characterized by high indexes of factor loadings, ranging most often from 0.47 to 0.78. Separate subscales of the MBRSQ-PL questionnaire were defined: (1) self-esteem of the body and its parts, (2) self-assessment of physical, (3) self-assessment of external appearance, (4) Negation of one’s own physical activity, (5) Self-assessment of health condition, (6) health anxiety, (7) fear of gaining weight, (8) neglecting health and appearance.

## 1. Introduction

Contemporary psychology and medical sciences indicate the necessity to use in the process of scientific research and clinical practice procedures consistent with the principles of evidence-based EBP in medicine, psychology, and psychotherapy [1]. In order to meet the applicable standards determining the procedures for conducting research, one should define and understand the body image as a multifactorial and multidimensional cognitive–emotional–behavioral structure [2]. Following the conceptualization of the body image recognized in the cognitive–behavioral literature, Cash constructed a tool that meets the above standards to measure and define various cognitive, emotional, and behavioral aspects of the body image. This tool is the Multidimensional Questionnaire of Relations to One’s Own Body (MBSRQ)—it is one of the most important and most commonly used tools for the multidimensional measurement of a person’s relationship to their body and appearance [2,3,4,5]. In many years of research, Cash and his team measured the spectrum of many factors describing the body image in terms of relation to the body and its experience, along with behavior towards the body and eating [2,6,7,8,9,10,11,12,13,14,15,16,17]. The widespread use of the MBRSQ in scientific research as a tool for measuring body image is reflected in numerous adaptations and validations of the MBRSQ ranging from German, Spanish, Greek, French, Chinese, Brazilian, Persian, and Malaysian [2,18,19,20,21,22,23]. The multifactorial structure of the MBSRQ in the Cash elaboration demonstrated sufficient and acceptable internal consistency and stability as well as strong construction validity in studies conducted on a large population of American men and women [2,24,25], as well as in clinical or quasi-clinical groups [16,26]. The MBRSQ questionnaire allows for a reliable, accurate, and critical analysis of the obtained indicators describing the relation to the body, taking into account the assessment of attitude towards physical appearance, physical fitness, and health [17,21]. Following the need for scientific verification of the variable body image and its disorders, one should strive to derealize the measurement of the variable based on numerous samples of the respondents. So far in Poland, there is no tool for measuring the variable body image that would have standardization or Polish norms adjusted to the population of both women and men. The available MBRSQ validation developed in Poland was based on a small population of 341 young women only aged 23 years and was published in 2015 by Anna Brytek-Matera and Radosław Rogoza [27]. However, the validation prepared by Britek and Rogoza, apart from the small number of young women surveyed, did not take into account the male population, nor did it propose to develop separate standards for the population of women and men. Taking into account the MBRSQ standardization studies carried out on numerous populations, especially women, the authors of this article conducted many years of research on a large population of healthy and age-matched Polish women and men in order to validate and standardize the MBRSQ along with the standards for the population of women and men, which are lacking in Poland. 

The aim of the article is to present the results of research aimed at collecting, analyzing, and presenting the Polish adaptation and standardization of the Multidimensional Body-Self Relationship Questionnaire—Appearance Scale (MBSRQ-AS) together with the results of a normalization scale for adult Polish men and women.

## 2. Materials and Methods

In the current Polish study, the psychometric properties and factor structure of the Polish version of MBSRQ-AS and the 69-item version were tested.

The Multidimensional Body-Self Relations Questionnaire (MBSRQ) by Thomas Cash [14,28] comprises 69 questions grouped into subscales clustered into three areas: self-assessment of the general appearance of the body and its individual parts—Appearance Evaluation (AE), Appearance Orientation (AO), and Body Areas Satisfaction (BASS); self-assessment of the health condition of the body—Health Evaluation (HE) and Health Orientation (HO); Illness Orientation (IO); Fitness Evaluation (FE), and Fitness Orientation (FO); and self-assessment of body weight and the level of fear of gaining weight—Overweight Preoccupation (OP) and Self-classified Weight (SCW). The respondents provide answers on the Likert scale containing the following possibilities: from 1 (“I strongly disagree”) to 5 (“I strongly agree”). The respondents, when making a self-report, assessed the level of their relationship to the body. 

Adaptation research on the Polish adaptation of the Multidimensional Body-Self Relations Questionnaire-Appearance Scales (MBSRQ-AS) was started in accordance with the procedures defined for cultural adaptation of the psychological test described by Hornowska and Paluchowski [29]. The consent of the MBRSQ author of the original version was obtained for its use in research. When starting work on the construction of the Polish version of the MBRSQ questionnaire, independent psychologists translated the original tool into Polish (the names of the questionnaire, 69 items of the questionnaire, and the names of individual 9 subscales as well as the content of items in these subscales).

Subsequently, English-fluent psychologists agreed on a common language version. Back translation was performed successively. Both versions were compared in order to check the accuracy of the translation [29]. The translations turned out to be very similar to the original tool.

The research was conducted from the years 2016 to 2021 in Poland, conducting research on the adult population in several cities. The study covered adults (men and women), studying, who had already completed their education and are working professionally (having various forms of employment) or are not currently working. All persons were informed that participation in the research was voluntary and anonymous.

The criteria for exclusion from the study were: declaring that the examined person had a diagnosis of eating disorders in their life history (medical diagnosis of anorexia, bulimia, compulsive overeating syndrome and treatment for the above-mentioned reason, diagnosis of mental disorders requiring (according to the respondents declared) various forms of treatment (in clinics, hospitals, and other health care facilities), people with body deformities due to various forms of physical disability within the body). Ultimately, the standardization sample consisted of 2688 respondents, including 1699 women and 1089 men.

Initially, confirmatory factor analysis was assumed to verify the factor structure of the tool in accordance with the original assumption of the author of the scale [14,28]. However, the fit indices reached unsatisfactory values, indicating that the structure of the tool may be different than its original assumption. For this reason, the next step was to use exploratory factor analysis. The number of factors and the value of factor loadings for individual items turned out to be satisfactory. The factors (subscales) of the questionnaire separated in this way were characterized by a satisfactory level of reliability. The aim of this article is to present the results of the research of the authors of the article, which were aimed at collecting, analyzing, and presenting Polish adaptation and standardization along with the standards for women and men. Multidimensional Body-Self Relations Questionnaire—Appearance Scales (MBSRQ-AS) in Polish Adults (women and men).

## 3. Results

### 3.1. Test Procedure and Characteristics of the Respondents Group

The study controlled: age of the respondents, sex and body mass index (BMI) level. For this reason, the research group consisted of women (N = 1699) and men (N = 1089) aged from 20 to 63 (M = 23.65; SD = 8.55), with a BMI level from 19 (M = 22.16; SD = 3.57) see Table 1 below.

Analysis of the results for the assessment of mean BMI and age shows that persons is within the normal range of body weight for the age of life.

### 3.2. Factor Structure of the Polish Version of the MBSRQ-PL Questionnaire

In order to investigate the psychometric properties of the adapted Polish tool, an exploratory factor analysis (EFA) was performed. Then, the reliability coefficients were calculated, and the descriptive statistics of individual subscales were checked. All analyses were performed with the use of two statistical packages: Statistica 10.0 and SPSS for Windows 23.0. Following the authors of the original version of the tool [30], exploratory factor analysis was performed using the principal components method, which was subjected to Varimax rotation with normalization, allowing the existence of a correlation between individual factors, but not excluding the lack of correlation and Kaiser normalization. The sampling compliance measures were as follows: KMO = 0.929, the Bartlett sphericity test result allowed to reject the unit matrix hypothesis (chi-square = 790.49; df = 2346; *p* < 0.001).

The results clearly indicated that this tool probably consists of subscales and does not constitute a uniform factor structure. This justified the analysis in this respect. 

The conducted exploratory factor analysis showed that both the Kaiser-Guttman criterion (loads above the value 1) and the results of the analysis of the “scree” plot justified the adoption of the eight-factor solution (see Figure 1). Based on the above analysis, eight factors were distinguished, explaining in total about 50% of the variance of the collected results (Table 2). The separated subscales are characterized by high indexes of factor loadings, ranging most often from 0.47 to 0.78. The principle was to obtain the greatest possible similarity of the values of the factor loadings presented in the English-language research of the team led by Cash. It was an additional criterion allowing for the final adoption of the eight-factor structure of the MBRSQ-PL questionnaire in Polish research.

After analyzing the eigenvalues plot and compiling the explained variance in terms of the identified factors, a decision was made to take into account the eight-factor structure of the tool. The adoption of such a construction of the tool was dictated, firstly, by its similarity to the original multivariate structure of the MBRSQ tool, as well as the fact that eight factors in total explain about 50% of the variance of the raw results collected. The table below contains a detailed list of the identified factors along with the specification of the factor loadings of the items classified to each factor. 

### 3.3. Characteristics of the Scales of the Polish Version of the MBRSQ-PL Questionnaire

The next stage of the research was an attempt to develop names for the selected factors of the Polish version of the MBRSQ-PL questionnaire. In the Polish adaptation of the questionnaire, there were differences in the strength of factor loadings and the number of items belonging (in accordance with the exploratory factor analysis carried out) to the eight distinguished factors. Therefore, it was necessary to redefine the original names of the identified factors for the Polish population. The list of factors and their Polish names are presented in the Table 3 below.

In the next step of analyzing the tool, separate subscales of the MBRSQ-PL questionnaire were defined.
(1)Self-esteem of the body and its parts (factor I; 12 items: 5, 11, 21, 30, 39, 61, 63, 64, 65, 66, 67, 69)—a scale describing the level of satisfaction or dissatisfaction with the body and its individual parts and body weight. High scorers are mostly positive and satisfied with the assessment of their own body and its parts. People with low scores generally have low satisfaction with their body and its individual parts and are dissatisfied with it.(2)Self-assessment of physical fitness (factor II; 10 items: 3, 4, 14, 24, 26, 35, 44, 51, 52, 53—a scale describing the level of satisfaction or dissatisfaction with one’s own physical activity and fitness. High achievers are satisfied with their fitness and are satisfied with the level of their own physical activity, they take care of their own physical condition. Low-scorers generally have low satisfaction with their own activity and fitness and are not fitness-oriented.(3)Self-assessment of external appearance (factor III; 7 items: 1, 2, 12, 13, 22, 31, 50)—a scale describing the level of focus on one’s own appearance. High scorers are overly focused on their appearance and how to improve it; low scorers are not focused on their own appearance and trying to constantly improve it.(4)Negation of one’s own physical activity (factor IV; 8 items: 6, 15, 16, 25, 33, 34, 38, 43—a scale describing the level of importance of one’s own physical activity in the life of the respondent. High achievers describe their physical activity as insignificant in everyday life, and they do not care about this aspect. In their behavior, they do not show interest in taking care of their own physical fitness. People with low scores generally attach great importance to their own activity and physical fitness, and in their lives, they engage in behaviors related to maintaining physical fitness.(5)Self-assessment of health condition (factor V; 8 items: 7, 9, 18, 19, 29, 46, 55, 56)—a scale describing the attitude of taking care of one’s own health. People who achieve high scores show a high level of care for their own health, consider health a significant value in life and pay attention to the symptoms of the disease flowing from the body. Low-scorers do not pay attention to the body’s symptoms of disease and health-threatening behavior, and health alone is not of great value to them.(6)Health anxiety (factor VI; 3 items: 17, 36, 45)—a scale describing the sense of lack of control over maintaining health condition. High scorers exhibit behaviors that indicate a lack of control over maintaining their own health. Persons with low scores show behaviors that indicate the retained control related to the pursuit of health and are aware of symptoms that indicate the health situation of their own body.(7)Fear of gaining weight (factor VII; 5 items: 10, 57, 58, 59, 60)—scale describing the level of fear of gaining weight. High scorers show a high level of fear of gaining weight and a tendency to restrictive behaviors such as dieting, fasting, etc. Low scorers show a low level of fear of gaining weight and a low tendency to restrict body and eating behavior.(8)Neglecting health and appearance (factor VIII; 4 items: 32, 37, 47, 49)—a scale describing the exposure of behaviors that indicate a lack of care for appearance and health. People who achieve high scores show an increased tendency to exhibit behaviors indicating a lack of interest in caring for their health, and they do not pay attention to their symptoms indicating a possible illness or their appearance. People with low results show a tendency to care for their health, and pay attention to symptoms flowing from the body, they also take care of their appearance.

### 3.4. Analysis of the Reliability and Consistency of the MBSRQ-PL Questionnaire

The next stage in the adaptation of the Polish MBRSQ-PL questionnaire was to check the reliability of the identified factors. For this purpose, a statistical analysis was performed to determine the level of Cronbach’s alpha reliability coefficients. The Table 4 below provides a detailed summary of the values obtained.

Due to the fact that the obtained values of Cronbach’s alpha coefficients exceeded the value of 0.4 (which is a satisfactory result, see Hornowska [31], it was concluded that MBSRQ -PL is characterized by satisfactory reliability. Cronbach’s α coefficients for individual subscales of the tool turned out to be satisfactory, i.e., they ranged from 0.659 to 0.900. The values of the reliability coefficients calculated for the original nine-factor structure of the MBRSQ, in the Cash [30] ranged from 0.70–0.900. Thus, the Polish version of the tool is characterized by a significant similarity in terms of reliability compared to the original version. 

In order to determine the internal consistency of the tool, the relationships between the factors of the MBSRQ-PL questionnaire were analyzed(Table 5).

The results indicate a satisfactory internal consistency of the identified factors of the MBSRQ-PL questionnaire. Therefore, it can be concluded that the multifaceted study of the body image (in the form of separate subscales) is an internally coherent construct. 

### 3.5. Descriptive Statistics of the Subscales of the MBRSQ-PL Questionnaire

In the presented research, the presentation of the results and, consequently, the sten standards of the questionnaire was undertaken, taking into account the sex of the respondents. The presentation of separate factors defining the areas indicated in eight factors (self-esteem of individual parts of the body, appearance, and perception of one’s own body and its physical condition and health) is the result of a specific and different attitude to the self-esteem of the body and appearance in women and men. The self-esteem of a man’s body will be subject to a slightly different evaluation system (in terms of norm and pathology) than in women. Epidemiological data show that women suffer from anorexia more often than men and they more often show a tendency to restrict eating behaviors [31,32,33], while men who are of the same age as women, in their behavior towards the body and eating, more often refer to the behavior, focused on increasing body muscles and physical activity. What will be identified with dissatisfaction for a woman due to the perception of her body weight, in her opinion as too high, in men may be related to the assessment of body as a normal body weight. Specific indices of body image in men differ from the severity of indices of normal body image in men (leanness–musculature). For this reason, the indicators and norms describing the specificity of the body image norm in men and women are different. For this reason, it seems justified to present separate results for women and men. Due to the above-mentioned differences between men and women in the ways of pursuing an idealized body appearance, sten norms have been developed separately for men and women. The Table 6, Table 7 and Table 8 below provide detailed information on this.

### 3.6. Polish Standardization of the MBRSQ-PL Questionnaire

Due to the conducted statistical analyses and the number of surveyed women (N = 1699) and men (N = 1089), it was possible to refer the raw results to the standardized sten scale. The Table 7 and Table 8 below provide detailed performance ranges. 

## 4. Discussion

In the presented research, the Polish adaptation of the Multidimensional Body Relations Questionnaire—Appearance Scale [30] was performed. The conducted confirmatory factor analysis, examining the original factor structure of the tool showed a complete lack of fit of the Polish population data. The conducted exploratory factor analysis allowed us to identify the structure of factors similar (though not the same) to the version of the original factor structure adopted by the authors.

In many adaptations of the MBSRQ questionnaire, (including the Brazilian, Spanish, Malaysian, Persian, and German versions) [19,20,21,22,34], the confirmatory factor analysis allowed us to accept the original set of factors. As mentioned earlier, in the Polish study, a similar direction of analysis turned out to be wrong. Exploratory factor analysis showed that in the Polish version of MBSRQ PL, the results were concentrated around eight factors (and not as originally assumed—seven). The size of the explained variance of the factors identified in this way was 50% and is comparable to the results of the validation of the MBSRQ AS test in the study by Brytek and Rogoza [27], where the explained variance was indicated at 52.57%.

The difference between the original set of factors and that identified in the Polish population is slight. The assumption of the researchers was to obtain the greatest possible similarity in the values of the system of factors presented in the English-language research of the team led by Cash. He indicates originally seven factors (main scales: Appearance Evaluation—AE; Appearance Orientation—AO; Fitness Evaluation FE; Fitness Orientation—FO; Health Evaluation HE; Health Orientation HO; Illness Orientation—IO) and three subscales (Body areas satisfaction scale—BASS; Overweight preoccupation—OWP; Self-classified weight—SCW).

The obtained structure of eight factors in the MBSRQ PL version indicates that factor I was made up of items identical in content in the original version of Cash to the AE (5, 11, 21, 30) and BAS (39, 61, 63–69) scales, describing body assessment and subjective assessment of a part of the body separately. In the MBSRQ PL version, items describing subjective experience and relations to the body and its parts make up one factor called: Self-esteem of the body and its part. 

Factor II (called Self-assessment of physical fitness) was formed by items that in the original version of Cash belonged to two scales: FE and FO. In the Polish version of MBSRQ Pl, factor II describes both the attitude towards self-assessment of one’s own physical fitness and the behaviors undertaken aimed at physical fitness. 

Factor III in MBSRQ-PL is called: the self-assessment of physical appearance, in the original version of Cash, included items belonging to one factor—the AO scale. This scale includes items that describe judgment, attitude to external appearance rather than experience, and relationship to one’s own body.

Factor IV in MBSRQ PL is called: the Negation of own physical activity scale, in the original version of MBSRQ Cash, contains items mainly belonging to the FO scale (six items) and included individual items belonging to other scales: FE (one item) and HO (one item). In connection with the above, we may consider factor IV MBSRQ Pl to be similar to factor FO in the original version. 

Due to the content of the items, factor V was called: the Self-Assessment of Health, which in the original version of the MBSRQ developed by Cash combines items belonging to the HO scale (four items), IO (three items), and HE scale (one item). This factor in the Polish version of MBSRQ PL, therefore, includes and combines the assessment of a person’s health condition, caring for it, and the perception of symptoms (indicators) of the disease in the context of noticing signs of disease from the body. 

Factor VI in MBSRQ PL called: health anxiety, includes three items that in the original version of Cash belong to the HE scale, which describes only the self-esteem of subjective emotions of anxiety and lack of control with regard to the assessment of health, disregarding the scope of self-esteem of undertaken actions and reactions to your drow condition and signs of disease (which include factor V). In the original version, IO, HE, and HO are separate scales. 

Factor VII in the Polish version of MBSRQ PL (fear of gaining weight) includes five items describing the fear of gaining weight and therefore focusing on weight control. In the original version, items in this factor such as 10, 57–60 belong to the OP and SCW scale, which separately describe the assessment of body weight and focus on body weight and anxiety related to weight gain.

Factor VIII was identified only in the Polish version of MBSRQ PL, it was named Neglect of health and appearance due to its content. Among the items included in this factor, four were originally composed of the following scales: AO and IO. In the Polish version of MBSRQ-PL, this factor consisted of four items, in the original version, the IO scale consisted of five items and constitute a separate factor.

## 5. Conclusions

In the presented research, the Polish adaptation of the Multidimensional Body-Self Relations Questionnaire-Appearance Scales [30] was performed. The MBRSQ-PL is characterized by fully satisfactory reliability for individual subscales. The values of the reliability coefficients obtained in the Polish adaptation of the questionnaire are similar to those obtained by the author of the tool [30]. In addition to the above-mentioned tool, the Sociocultural Attitudes Questionnaire towards the Body Appearance (SATAQ 3) is also used in Poland, but it refers to the measurement of sociocultural attitudes towards the appearance and not to the assessment of the relationship to the body and the experience of the body, as by MBSRQ [35]. In terms of measuring variables related to body image, the Body Esteem Scale by S. Franzoi and S. Shields in Polish adaptation by M. Lipowska and M. Lipowski is also available [36]. However, the questionnaire deals with a slightly different aspect of the description of body image, without reference to the relationship to the body. So, this is a limited description of the body image variables.

Summing up, the presented research has shown that MBRSQ-PL is a tool with satisfactory psychometric properties. The obtained results confirm the validity application of the Multidimensional Body-Self Relations Questionnaire—Appearance Scales—PL in research with a group consisting of people of Polish nationality. This means that MBRSQ-PL is a valuable source of information on body image perception, relationship, and experience of the body and its physicality and thus can be an alternative to questionnaires available in Poland examining this area of variables. The questionnaire may be a useful tool due to the different norms available for women and men for the diagnosis of a wide spectrum of indicators of the relationship to the body in a psychological diagnosis. The results may also be helpful in the development of psychoeducational and preventive programs related to a healthy perception of the body image in a wide range of ages. It would be worth supplementing the presented results of the Polish adaptation of the tool with further analyses, e.g., by checking the correlation between the tool and other questionnaires used in Poland in the field of measuring body image perception.

## 6. Limitation

The presented research results have some limitations. According to the authors of the study, the most important ones include issues such as failure to define validity. The conducted analysis does not take into account the comparisons of the results of the MBRSQ-AS and another tool, which measures indicators from the area of the body image. However, the authors believe that the study used a questionnaire that has been well-established in global research for many years and has been the subject of many studies in the international arena. Another identifiable limitation of the presented research is including only healthy people in the research group. There are disorders, (e.g., eating disorders) that are characterized, among others, by body image disturbance. In the presented research, the presence of such disorders was one of the criteria for excluding from the research group. Thus, the presented normalization refers only to the determination of the areas of one’s own body in a group of healthy people. Another observed limitation is the fact that the studies were conducted only on the population of Polish adults. Although such a condition was assumed by the authors of this study, it should be clear that the calculated norms apply only to the Polish population.

## Figures and Tables

**Figure 1 ijerph-19-06097-f001:**
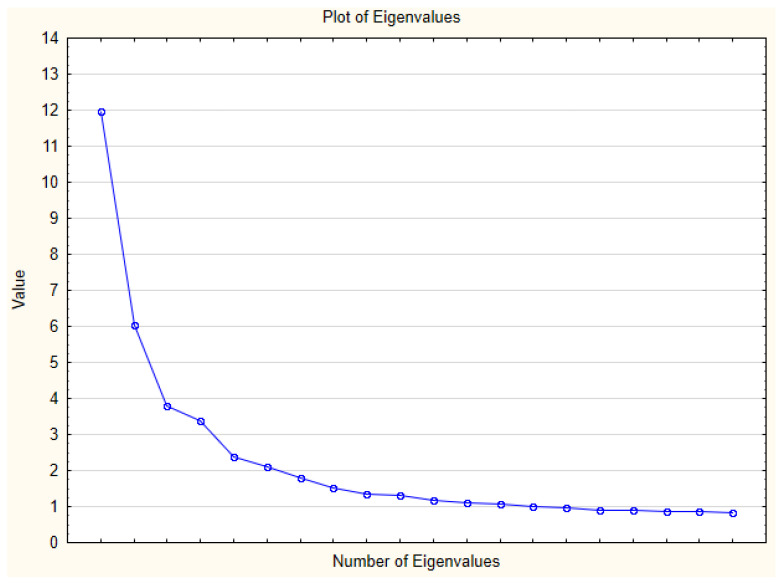
Plot of Eigenvalues of factors identifies in the MBRSQ-PL questionnaire.

**Table 1 ijerph-19-06097-t001:** Descriptive statistics of total results for individual subscales.

Variables	N	M	Me	Min.	Max.	SD
Men						
Age	1089	24.36	23.00	20.00	66.00	7.86
BMI	1089	22.57	21.54	20.00	41.96	4.82
Women						
Age	1699	24.06	25.00	20.00	68.00	7.52
BMI	1699	21.37	20.73	19.00	35.11	3.42

**Table 2 ijerph-19-06097-t002:** Exploratory factor analysis of the MBSRQ-PL questionnaire.

Number Item (According to the Original MBRSQ Tool)	F1	F2	F3	F4	F5	F6	F7	F8
5 My body is sexually attractive	0.73							
11 I like my appearance the way it is	0.76							
21 Most people would think I look good	0.59							
30 I like the way I look without clothes	0.77							
39 I like the way the clothes sit on top of me	0.66							
61 Face (facial features, complexion)	0.64							
63 Lower torso (buttocks, hips, thighs, legs)	0.6							
64 Middle Torso (Waist, Abdomen)	0.61							
65 Upper torso (breasts, shoulders, arms)	0.66							
66 Musculature	0.52							
67 Weight	0.59							
69 Overall appearance	0.84							
3 I would have passed most fitness tests		0.73						
4 It is important that I have above-average physical strength		0.63						
14 I have good physical endurance		0.72						
24 I find it easy to learn new sports skills		0.73						
26 I do various things to increase my physical strength		0.67						
35 I am working on improving my endurance		0.63						
44 I try to be physically active		0.64						
51 I have very good coordination		0.65						
52 I know a lot about fitness		0.71						
53 I do sports regularly throughout the year		0.66						
1 Before leaving the house, I always pay attention to what I look like			0.73					
2 I am careful about buying clothes that will keep me looking my best			0.68					
12 I check myself in the mirror whenever I can			0.67					
13 Before I leave home, I usually take a long time to prepare			0.68					
22 It is important that she always looks good			0.75					
31 I am embarrassed if I am not groomed properly			0.49					
50 I always try to improve my appearance			0.7					
6 I do not do exercise on a regular basis				−0.68				
15 Playing sports is not important to me				−0.47				
16 I am not actively doing anything to keep in good shape				−0.75				
25 Being in good physical shape is not a big priority in my life				−0.68				
33 I am not very good at sports and games				−0.49				
34 I rarely think about my fitness				−0.64				
38 I do not make any special efforts to eat a balanced and nutritious diet				−0.45				
43 I do not care about improving my sports prowess				−0.65				
7 I am in control of my health					0.47			
9 I consciously developed a healthy lifestyle					0.46			
18 Good health is one of the most important things in my life					0.57			
19 I do not do anything that I know would endanger my health					0.57			
29 I often read health books and magazines.					0.5			
46 I pay close attention to my body for any signs of illness					0.5			
55 I am fully aware of the small changes in my health					0.5			
56 At the first sign of my illness, I seek medical advice					0.55			
17 My health is something that gets better or worse unexpectedly						−0.62		
36 Overnight, I never know how my body will feel						−0.47		
45 I often feel prone to illness						−0.74		
10 I am constantly worried that I am or will get fat							−0.55	
57 I am on a slimming diet							−0.47	
58 I tried to lose weight by fasting and following restrictive diets							−0.49	
59 I think I weigh too much							−0.75	
60 Looking at me, most people would think I am fat							−0.73	
32 I usually put on what I have at hand, not caring what it looks like								0.51
37 When I am sick, I do not pay much attention to my symptoms								0.74
47 If I get a cold or the flu, I ignore it and go back to normal								0.58
49 I never think about my appearance								0.68

**Table 3 ijerph-19-06097-t003:** A list of the names of the distinguished factors in the MBSRQ-PL questionnaire.

Number	Name
Factor 1 (F1)	Self-esteem of the body and its parts
Factor 2 (F2)	Self-assessment of physical fitness
Factor 3 (F3)	Self-assessment of external appearance
Factor 4 (F4)	Negation of one’s own physical activity
Factor 5 (F5)	Self-assessment of health condition
Factor 6 (F6)	Health anxiety
Factor 7 (F7)	Fear of gaining weight
Factor 8 (F8)	Neglecting health and appearance

**Table 4 ijerph-19-06097-t004:** Cronbach’s alpha reliability coefficients for subscales of the MBRSQ-PL questionnaire.

Factors (Subscales)	Cronbach’s α
Self-esteem of the body and its parts	0.900
Self-assessment of physical fitness	0.909
Self-assessment of external appearance	0.820
Negation of one’s own physical activity	0.819
Self-assessment of health condition	0.733
Health anxiety	0.677
Fear of gaining weight	0.733
Neglecting health and appearance	0.659

**Table 5 ijerph-19-06097-t005:** Correlation matrix between the subscales of the MBSRQ-PL questionnaire.

	F1	F2	F3	F4	F5	F6	F7
Self-esteem of the body and its parts (F1)	-						
Self-assessment of physical fitness (F2)	0.400 ***						
Self-assessment of external appearance (F3)	−0.018	0.136 ***					
Negation of one’s own physical activity (F4)	−0.161 ***	−0.510 ***	−0.083 ***				
Self-assessment of health condition (F5)	0.330 ***	0.417 ***	0.191 ***	−0.260 ***			
Health anxiety (F6)	−0.207 ***	−0.155 ***	0.099 ***	0.286 ***	−0.062 ***		
Fear of gaining weight (F7)	−0.497 ***	−0.113 ***	0.203 ***	0.047 *	−0.054 **	0.198 ***	
Neglecting health and appearance (F8)	−0.044 *	−0.030	−0.220 ***	0.275 ***	−0.124 ***	0.217 ***	0.040 *

*** *p* < 0.001; ** *p* < 0.01; * *p* < 0.05.

**Table 6 ijerph-19-06097-t006:** Descriptive statistics of total results for individual subscales for the variables included in the MBRSQ-PL questionnaire in the research group.

Variables	N	M	Me	Min.	Max.	SD
Men						
(1) Self-esteem of the body and its parts	1089	38.79	40.00	12.00	60.00	9.84
(2) Self-assessment of physical fitness	1089	31.62	32.00	10.00	50.00	9.33
(3) Self-assessment of external appearance	1089	23.02	23.00	7.00	35.00	5.71
(4) Negation of one’s own physical activity	1089	22.43	23.00	8.00	40.00	7.31
(5) Self-assessment of health condition	1089	22.91	23.00	8.00	40.00	5.77
(6) Health anxiety	1089	7.8	7.00	3.00	15.00	3.14
(7) Fear of gaining weight	1089	12.11	12.00	5.00	24.00	3.77
(8) Neglecting health and appearance	1089	10.52	10.00	4.00	20.00	3.55
Women						
(1) Self-esteem of the body and its parts	1699	37.31	39.00	12.00	60.00	10.64
(2) Self-assessment of physical fitness	1699	30.28	30.00	10.00	50.00	9.32
(3) Self-assessment of external appearance	1699	24.75	25.00	7.00	35.00	5.52
(4) Negation of one’s own physical activity	1699	23.51	23.00	8.00	40.00	7.15
(5) Self-assessment of health condition	1699	24.06	24.00	8.00	40.00	5.50
(6) Health anxiety	1699	8.75	9.00	3.00	15.00	298
(7) Fear of gaining weight	1699	13.59	13.00	2.00	25.00	3.95
(8) Neglecting health and appearance	1699	10.64	10.00	4.00	20.00	3.77

**Table 7 ijerph-19-06097-t007:** Standards of the MBSRQ-PL questionnaire in the group of women.

Sten	F1	F2	F3	F4	F5	F6	F7	F8
1	<18	<13	<15	<10	<14	<3	<6	<4
2	19–24	14–18	16–17	14–11	15–17	5–4	8–7	5
3	25–29	19–23	18–20	15–18	18–19	6	10–9	7–6
4	30–34	24–27	21–23	19–21	20–22	8–7	12–11	9–8
5	35–39	28–32	24–26	22–25	23–25	9	13–14	11–10
6	40–45	33–37	27–28	26–28	26–28	10	15–16	13–12
7	46–50	38–41	29–31	29–32	29–30	12–11	17–18	14–15
8	51–55	42–46	32–34	33–36	31–33	13	19–20	16–17
9				37–39	34–36		21–22	18–19
10	>56	>47	>35	40	>37	>14	>23	20

Legend: F1—self-esteem of the body and its parts; F2—self-assessment of physical fitness; F3—self-assessment of external appearance; F4—negation of own physical activity—physical exertion; F5—self-assessment of health condition; F6—health anxiety; F7—fear of gaining weight; F8—neglecting health and appearance.

**Table 8 ijerph-19-06097-t008:** Standards of the MBSRQ -PL questionnaire in the group of men.

Sten	F1	F2	F3	F4	F5	F6	F7	F8
1	<21	<15	<13	<9	<12	<3	<5	<4
2	22–26	16–19	14–15	13–10	13–15		7–6	6–5
3	27–31	20–24	16–18	14–16	16–18	5–4	9–8	7
4	32–36	25–29	19–21	17–20	19–21	7–6	11–10	9–8
5	37–41	30–33	22–24	21–24	22–24	8	13–12	11–10
6	42–46	34–38	25–27	25–27	25–27	10–9	14	13–12
7	47–51	39–43	28–30	28–31	28–30	11	15–16	14
8	52–56	44–47	31–33	32–35	31–33	13–12	17–18	15–16
9				36–38	34–35	14	19–20	17–18
10	>57	>48	>34	>39	>36	15	>21	>19

Legend: F1—self-esteem of the body and its parts; F2—self-assessment of physical fitness; F3—self-assessment of external appearance; F4—negation of own physical activity—physical exertion; F5—self-assessment of health condition; F6—health anxiety; F7—fear of gaining weight; F8—neglecting health and appearance.

## Data Availability

Not applicable.

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
