# Peer review of "Factor Structure of the Polish Version of Multidimensional Body-Self Relations Questionnaire-Appearance Scales (MBSRQ-PL)"

_ijerph, 2022, doi:10.3390/ijerph19106097_

Round 1

Reviewer 1 Report

Dear authors,

This study explores an important issue and contains interesting findings. However, there are some suggestions for improvement: 

  • According to the contents of the article the phrase “factor structure of the Polish version of the MBSRQ” is more appropriate than “MBSRQ; Polish adaptation” for the title.
  • The section on Materials and Methods is mixed with the Results section. They should be separated.
  • The description of the original measure to be adapted should be included the Materials and Methods section.
  • Line 20 – The number of the first scale is missing.
  • Line 124 – An abbreviation “BMI” should be defined before being introduced to the text on page 3.
  • Line 126 – The number of the table is missed in the sentence.
  • Lines 140-151 – Description of the data analysis is moved to the results section. They should be separated and described in the proper sections.
  • Line 149 – Pilot study is mentioned here. It is not clear what was the pilot study and what was the main one in the presented research. It should be explained.
  • Line 259 – In the note to the table the last asterisk should be moved before “p”.
  • Lines 386 – 401 – The tables 7 and 8 and their description should be moved to the Results section.

Author Response

Thank you very much for your insightful suggestions. We took into account all the reviewer's requests.

  • According to the contents of the article the phrase “factor structure of the Polish version of the MBSRQ” is more appropriate than “MBSRQ; Polish adaptation” for the title. (review included)
  • The section on Materials and Methods is mixed with the Results section. They should be separated. (review included)
  • The description of the original measure to be adapted should be included the Materials and Methods section. (review included)
  • Line 20 – The number of the first scale is missing. (review included)
  • Line 124 – An abbreviation “BMI” should be defined before being introduced to the text on page 3. (review included)
  • Line 126 – The number of the table is missed in the sentence. (review included)
  • Lines 140-151 – Description of the data analysis is moved to the results section. They should be separated and described in the proper sections. (review included)
  • Line 149 – Pilot study is mentioned here. It is not clear what was the pilot study and what was the main one in the presented research. It should be explained. (review included, a clerical mistake)
  • Line 259 – In the note to the table the last asterisk should be moved before “p”. (review included)
  • Lines 386 – 401 – The tables 7 and 8 and their description should be moved to the Results section. (review included)

Reviewer 2 Report

Dear authors:

First of all I congratulate you on your excellent work. Here are my recommendations for improving your manuscript:
- Include bibliographical references in the abstract. In my opinion, it is not necessary to cite them in this section, but if you do, you should do so in order of appearance.
- Citations that are consecutive should be numbered as follows (6-19) on lines 40-41 y (20-25) on line 44. 
-
- The material and methods section is not completed, as it is the text of the template provided by the publisher. Without this section the research cannot be fully assessed.
- Line 115, the term voivodeships must be defined.
- Lines 132-137 should be included in the material and methods section.
- They should include in the discussion section a sub-section on strengths and limitations of their research.
- In the conclusions section, they should provide a tight and concise conclusion, not so extensive, and they should avoid introducing tables in this section.

They should improve the manuscript as there is a major editing error.

Best regards.

Author Response

Thank youvery much for your valuable comments. The requests were accepted

- Include bibliographical references in the abstract. In my opinion, it is not necessary to cite them in this section, but if you do, you should do so in order of appearance. (we give up citing)
- Citations that are consecutive should be numbered as follows (6-19) on lines 40-41 y (20-25) on line 44.  (review included)
- The material and methods section is not completed, as it is the text of the template provided by the publisher. Without this section the research cannot be fully assessed. (review included)
- Line 115, the term voivodeships must be defined. (review included, we have removed this term,)
- Lines 132-137 should be included in the material and methods section. (review included)
- They should include in the discussion section a sub-section on strengths and limitations of their research. (strenghts and limitation are included in the summary section )
- In the conclusions section, they should provide a tight and concise conclusion, not so extensive, and they should avoid introducing tables in this section. (review included)

Round 2

Reviewer 2 Report

Dear authors:

Thank you for carrying out the review of your manuscript. Here are some suggestions for improving your article:
- In the discussion section you should include a sub-section in which you develop the limitations of your work.
- In line 38, you should cite references correctly (2-5).
- However, in line 40, the references given (6-19) seem too many citations to explain this concept.
- In my view, you should expand the discussion section, as it is short.

Good luck with the publication.

Author Response

Thank you for your comments, all of them have been approved by the authors